# Experimental Model for Study of Thickness Effect on Flexural Fatigue Life of Macro-Synthetic-Fiber-Reinforced Concretes



**Mohammad Daneshfar [1], Abolfazl Hassani [1], Mohammad Reza Mohammad Aliha [2], Tomasz Sadowski [3],\* and Arastoo Karimi [4]**

1   School of Civil and Environmental Engineering, Tarbiat Modares University, Tehran 14115-111, Iran
2   Welding and Joining Research Center, School of Industrial Engineering, Iran University of Science and Technology (IUST), Narmak, Tehran 16846-13114, Iran
3   Department of Solid Mechanics, The Lublin University of Technology, Nadbystrzycka 40 Str., 20-216 Lublin, Poland
4   Politecnico di Torino, Corso Duca degli Abruzzi, 24, 10129 Torino, Italy
\*   Correspondence: t.sadowski@pollub.pl

**Abstract:** As one of the most widely used building materials, concrete has a dominantly brittle or quasi-brittle behavior. Adding fibers to concrete affects its ductility behavior as well as some mechanical properties. Finding the relationship between the addition of fibers and the change in thickness of laboratory test samples made of concrete can help in designing the optimal thickness of real concrete layers (especially concrete pavements) to withstand dynamic loads. The purpose of this research is to provide an experimental model for investigating the effect of concrete specimen size, or the thickness effect of concrete sample, on the fatigue life of concrete. Accordingly, several concrete beams with three thicknesses (80, 100 and 150 mm), constant width, and two lengths (120 mm and 450 mm) were manufactured with fiber percentages of 0 and 4% by fraction volume. The employed fiber was twisted macro synthetic fiber. After curing for 28 days, the samples were subjected to fatigue loading at three stress levels until the onset of failure and cracking stage. Here, the experimental model of the relationship between the number of loading cycles, the stress level and the thickness of the sample is presented. The results show that increasing the specimen thickness and fiber content can enhance the fatigue life of concrete up to 68%.

**Keywords:** concrete; twisted fiber; flexural fatigue life; beam specimen; thickness effect

## 1. Introduction

Various types of fibers are used to produce fiber-reinforced concrete, including glass, polymer, carbon and steel [1]. Research has been conducted to investigate the effect of fibers on concrete, such as mechanical properties, fatigue life and durability, which indicate the positive effect of fibers on concrete performance [2–7]. In the present research, macro-synthetic polymer fibers were used. Some of the consequences of applying macro-synthetic fibers in concrete include reduced shrinkage of fresh and hardened concrete, increased ductility, increased strength against fatigue stresses, increased durability and lifetime of concrete, improved concrete mechanical properties (tensile strength, flexural strength, etc.), control of secondary/thermal cracks of concrete, preventing the in-depth propagation of cracks, post-cracking charge ability and reduced permeability against chloride and sulfate ions [8].

Zeybek et al. evaluated the performance of fiber-reinforced concrete produced with steel fibers extracted from waste tires. The experimental study was carried out to explore the effect of fiber content on the fresh and hardened state of the concrete. Compression, splitting tensile and flexure tests were carried out to observe the performance of the concrete with tire-recycled steel fibers with ratios of 1%, 2% and 3%. Improvement was shown in the mechanical properties of the concrete with the increased volume fraction of the steel

fiber. However, a significant reduction in workability was observed after the addition of 2% steel fibers. Therefore, they recommended utilizing 2% tire-recycled steel fibers in practical applications [9]. Aliha et al. [10], investigated the fracture toughness of cement concrete mixture reinforced with synthetic forta fibers under mode I and III fracture deformations. They showed that addition of such fibers can significantly enhance the post peak failure rupture resistance of plain concrete.

Other published research concerns the strength, performance and cracking resistance of concrete mixtures made of different additives (such as natural, synthetic and industrial fibers or waste and recycled materials) [11–24]. According to these studies, the type and shape of additive, size of additive (e.g., length or diameter of fibers or gradation of granules), percentages of additives and fiber dosages relative to the volume percentage of the total mass or volume of the concrete mixture and also the environmental conditions can noticeably affect the strength and fracture resistance of concretes. Depending on the type and content of additive in the mixture of concrete, the strength and mechanical properties of asphalt mixtures can be either reduced or enhanced [25–36]. Bawa et al. investigated the flexural fatigue life of self-compacting concrete with different proportions of steel fiber (SF) and polypropylene fiber (PPF) reinforcement with an experimental program and reported that the gradual replacement of SF with PPF in the mixes was found to significantly reduce the variability in the distribution of fatigue life. A maximum increase of 62% in the shape parameter for the fatigue life of a mix with 100% PPF, as with a mix containing 100% SF, was observed at a stress level of 0.85 [37]. Lu et al. conducted an experimental study on tensile fatigue strength of steel fiber reinforced concrete to show that the fatigue life of specimens were in good agreement with the Weibull distribution. Thus, a series of fatigue life equations under different survival probabilities were established, as well as the ultimate fatigue strength equation containing two parameters—the ratios of maximum and minimum stress to tensile strength [38]. Ahmad et al. indicated that polypropylene fiber improved the mechanical strength and durability of concrete (particularly tensile capacity) but decreased the flowability of concrete. The optimum dose is important, as a higher dose adversely affects strength and durability due to a lack of flowability. Scanning electronic microscopy results indicate that the polypropylene fibers restrict the propagation of cracks, which improves the strength and durability of concrete [39]. Chee Keong Lau et al. investigated fatigue performance of steel fiber reinforced rigid concrete pavement and reported that the addition of fibers at 0.5% volume fraction in concrete improved the fatigue life by at least 135% and reduced the energy dissipated per cycle by 74%. As the volume fraction of fibers increased, it was found that the fatigue life of rigid pavements improved [40]. Xu et al. studied the fatigue behavior of steel fiber reinforced concrete composite girders under high cycle negative bending action. Xu et al. reported that the fatigue load cycles in tests resulted in about 10–30% reduction of girders' global bending stiffness without interlay bonding, and the effect of SFRC on it was not obvious [41]. Chong Zhang conducted an experiment on the fatigue behavior of steel fiber reinforced high-strength concrete under different stress levels and showed that stress level significantly influenced the fatigue life of SFRHSC beams and the fatigue behavior of the beams was mainly determined by the tensile reinforcement [42]. Qu et al. conducted a study on flexural fatigue performance of steel fiber reinforced expanded-shales lightweight concrete superposed beams with initial static-load cracks and reported that with the increase of concrete depth and the volume fraction of steel fiber, the fatigue life of the test beams was prolonged with three altered failure modes due to the crush of conventional concrete in the compression zone and/or the fracture of the tensile rebar. The failure pattern could be more ductile by prevention of fatigue fracture by the longitudinal tensile rebar when the volume fraction of steel fiber was 1.6% and crack growth, concrete strain were reduced in the compression zone. The fatigue life of test beams was sensitive to the upper-limit of the fatigue load. A short fatigue life appeared from the higher stress level and larger stress amplitude of the longitudinal rebar due to the higher upper-limit of the fatigue load [43]. Alberti et al. studied self-compacting fiber-reinforced concrete by combining polyolefin

and metal fibers, and reported that the combination had higher performance in terms of rupture toughness and flexure than the use of a single type of fiber [44]. Jamshidi et al. reviewed the literature on the use of hybrid fibers in concrete and reported that combining various types of fibers would yield better results in terms of concrete toughness and energy absorption [45]. Eswari conducted an experiment on the flexural performance of hybrid fiber-reinforced concrete and evaluated the effect of different contents of polymer and steel fibers on the flexural strength and performance of the fiber-reinforced concrete samples. Thus, he reported that adding fibers could improve them in the evaluated parameters compared to conventional concrete [46]. Singh et al. studied the flexural strength and toughness of fiber-reinforced concrete with different percentages of polypropylene and steel fibers with a total of 1%. Results showed that combining 75% of the steel fibers and 25% of polypropylene fibers yielded better results in terms of compressive strength, flexural strength and flexural toughness [47]. Bedi et al. investigated the flexural fatigue lifetime of polymer polypropylene concrete at different levels of stress using numerical analyses and obtained parameters of the probabilistic models. They performed the flexural fatigue test on fiber-reinforced concrete samples with polypropylene fiber contents of 0.5%, 1% and 2%. Accordingly, they reported that the increased fiber content led to increased fatigue lifetime of the concrete samples [48]. Jiabiao et al. experimented with the cracking behavior and toughness of fiber-reinforced concrete as well as the advantages and features of adding synthetic fibers to concrete. Adding micro-fibers by 0.1% of the concrete's volume affected plastic shrinkage. In addition, adding micro-fibers by 1% of the concrete's volume affected the flexural toughness, impact strength and fatigue strength of the hardened concrete. Furthermore, the equivalent flexural strength was measured for post-cracking strength in various samples [49]. Bordelon and Roesler studied fiber reinforced concrete with steel, synthetic and steel mesh fibers, and reported that the use of fibers led to increased bearing capacity and reduced thickness of concrete pavement [50]. Kreiger conducted a study on a model to explain mode I rupture of high-performance steel fiber-reinforced concrete and showed that increasing the fiber percentage led to increased fracture energy. Additionally, by increasing the span length to depth ratio of the beam, the maximum rupture force was reduced [51].

The size effect of samples on fatigue fracture response was analyzed in several papers. Bazant et al. investigated this effect in fiber reinforced concrete and showed that the sample's fatigue life significantly depended on its size [52]. In 2006, Roesler investigated the fatigue strength of concrete pavement. The large-scaled concrete slab fatigue test in the laboratory showed that the fatigue life of concrete slab was much higher than that of concrete beam and depended on slab geometry, thickness, loading, concrete materials and boundary conditions. In addition, the flexural capacity of the concrete slab was 1.3–3.5 times more than the concrete beam samples [53]. Goel et al. investigated the fatigue life of conventional self-compacting metal fiber-reinforced concrete beams, and via providing Weibull distribution parameters, represented that the fatigue life of the conventional self-compacting metal fiber-reinforced concrete exhibited better performance than the standard concrete reinforced by fibers that had been vibrated conventionally [54]. Zhang et al. studied the effect of size on the flexural fatigue of concrete. By investigating the S-N diagrams (where S is the ratio of stress to maximum bending stress and N is several load cycles), they showed a strong dependence of fatigue life on the sample's size. Further, they reported that increasing the thickness and length of the beams led to reduced fatigue life [55]. More recently, Hoseini et al. [56,57], studied the effect of coarse aggregate content and wavy steel fiber percentages on mechanical properties including compressive strength, tensile strength, pure modes I and III fracture toughness and mixed mode I/III fracture toughness of self-compacting steel fiber reinforced concrete samples. They used single edge notched bend (SENB) beam and edge notched disc bend (ENDB) samples in their experimental research program. In other similar work, Ghasemi and co-workers [58] also investigated the influence of maximum nominal aggregate size on fracture parameters of self-compacting steel fiber reinforced concretes. Using edge notched

prismatic beam subjected to three-point bending, Pan et al. [59] experimentally studied the influence of steel fiber volume percentage and its length on the crack propagation speed of self-compacting fiber reinforced concrete. They used two types of steel fibers with lengths of 13 and 30 mm and three fiber volume ratios of 0.51%, 0.77% and 1.23%. Four loading rates (i.e., 2.66 m/s, 1.77 m/s, 22 mm/s and 2.2 µm/s) were used in their work to conduct the experiments. Digital image correlation (DIC) method and strain gages attached in different locations relative to the pre-notch were utilized for determining the velocity of growing crack in the concrete beam. Zeng et al. [60] reinforced the ultra-high performance concrete (UHPC) plates using fiber-reinforced polymer (FRP) grids. The influence of fiber type, fiber length and fiber content on the flexural strength value was studied in their work by performing 26 samples loaded by three-point bend setup. They also analyzed the microstructure of broken samples via scanning electron microscopy (SEM) and performed an analysis between strength and cost of the manufactured concrete samples [60]. Other testing samples and reinforcing fibers have also been investigated in recent years on mechanical properties and fracture toughness of concrete mixtures [61–64].

In most of the studies, the concrete sample's thickness is increased along with the increase in the beam's length; however, in the present work, only the thickness of the beam samples, with and without fibers, was changed and other dimensions of the samples were kept constant to investigate merely the effect of increased thickness. Accordingly, the size effect of the macro-synthetic fiber-reinforced concrete sample at different thicknesses was assessed via fatigue life variations. The inter-twisted fibers were added to the concrete mixture by 0.4 vol.%. The experimental program was focused on assessment of influence of two effects on the fatigue life of concrete samples, i.e., (1) addition of macro-synthetic fibers, and (2) thickness effect of the specimens and presentation of experimental models.

## 2. Materials and Methods

### 2.1. Test Variables

To evaluate the flexural fatigue life of concrete beams, the concrete mixture was designed based on the ACI 211 Standard [65]. All the concrete samples were made with the same mix designs as below:

(1)  Without addition of any reinforcement,
(2)  With 0.4 volume percentage of twisted fibers (Figure 1) used for the fibrous mixtures.

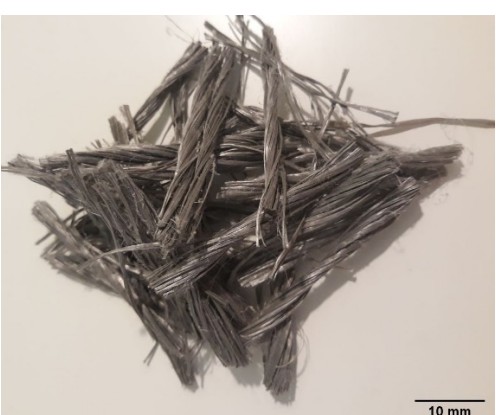

**Figure 1.** Twisted fibers used in the manufacturing of fiber-reinforced concrete.

In this research, concrete mix designs were coarse aggregate 880 kg/m$^3$, fine aggregate 789 kg/m$^3$, cement 442 kg/m$^3$, water 199 kg/m$^3$, superplasticizer 2.2 kg/m$^3$ and fibers with dose of 3.6 kg/m$^3$. The concrete mix design is described in Table 1. Figure 1 presents the fibers used in the present research [66].

**Table 1.** Concrete mix design.

| Materials | Fine Aggregate | Coarse Aggregate | Cement | Water | Fiber | Superplasticizer |
|---|---|---|---|---|---|---|
| Weight per unit volume (kg/m$^3$) | 789 | 880 | 442 | 199 | 0, 3.6 | 2.2 |

### 2.2. Materials and Specimen Preparation

The physical properties of siliceous aggregates are shown in Table 2.

**Table 2.** Aggregate properties.

| | |
|---|---|
| Nominal maximum aggregate size (mm) | 19 |
| Bulk specific gravity, g/cm$^3$ (ASTM C127) | 2.59 |
| Absorption fine aggregate/% (ASTM C127) | 2.4 |
| Absorption coarse aggregate/% (ASTM C127) | 1.2 |
| Los Angeles abrasion loss/% (ASTM C131) | 11.3 |
| Two fractured faces/% (ASTM D5821) | 93 |
| Soundness (ASTM C88) | 1.7 |
| Sand equivalent (ASTM D2419) | 70 |

The same gradation was selected for all the designed mix proportions. Moreover, Type I cement was used in this research. The chemical compositions and the mechanical and physical properties of this cement are given in Tables 3 and 4.

**Table 3.** The chemical composition of Type I Portland cement.

| Oxide | CaO | SiO$_2$ | Fe$_2$O$_3$ | SO$_3$ | Al$_2$O$_3$ | MgO | Na$_2$O | K$_2$O | LOI | TiO$_2$ | MnO |
|---|---|---|---|---|---|---|---|---|---|---|---|
| (%) | 64.73 | 19.98 | 4.11 | 3.79 | 3.50 | 2.07 | 0.15 | 0.63 | 0.35 | 0.27 | 0.20 |

**Table 4.** The compressive strength of Type I Portland cement.

| Days | Compressive Strength (MPa) |
|---|---|
| 3 | 22 |
| 7 | 26 |
| 28 | 35 |

To evaluate the flexural fatigue life of the fiber-reinforced concrete, cement was mixed with sands, gravel and fibers, and then the water was mixed with superplasticizer accordingly. Several rectangular beam specimens with and without twisted fiber were manufactured. According to the standard, the concrete samples were kept in the water pond for 28 days. Then, to obtain the stress levels in the fatigue test, the bending strength of at least three samples from each of the dimensions of the concrete samples was obtained and averaged. Table 5 shows the geometrical properties of the prepared specimens.

**Table 5.** Specification of tested specimens.

| Specimen No. | Shape of Fiber | Fiber Volume Fraction (%) | Specimen Size (mm) |
|---|---|---|---|
| D1 | Twisted | 0.4 | 80 × 20 × 450 |
| D2 | Twisted | 0.4 | 100 × 120 × 450 |
| D3 | Twisted | 0.4 | 150 × 120 × 450 |
| N1 | - | 0 | 80 × 120 × 450 |
| N2 | - | 0 | 100 × 120 × 450 |
| N3 | - | 0 | 150 × 120 × 450 |

### 2.3. Experimental Settings and Measurements

The UTM (Universal Testing Machine) with 100 ton capacity was used to determine flexural fatigue of the manufactured concrete samples. The tests were performed with constant sinusoidal loading at the frequency of 10 Hz. The input information for the test included

- Loading curve shape,
- Minimum and maximum loading values,
- Loading frequency,
- The maximum number of loadings.

To measure the values of stress levels, first, the average flexural strength of samples had to be determined; then, the stress levels could be calculated from the obtained results. Table 6 shows the input data for the UTM device.

**Table 6.** Input data for concrete samples with different thicknesses.

| Specimen | S | $P_{Max}$ (Average) | PxS | Amplitude | Bios |
|---|---|---|---|---|---|
| D1 | 0.7 | 12,211.5 | 8548.05 | 2137.01 | 6411.04 |
|    | 0.8 | 12,211.5 | 9769.2 | 2442.3 | 7326.9 |
|    | 0.9 | 12,211.5 | 10,990.35 | 2747.59 | 8242.76 |
| D2 | 0.7 | 17,527.2 | 12,269.04 | 3067.26 | 9201.78 |
|    | 0.8 | 17,527.2 | 14,021.76 | 3505.44 | 10,516.32 |
|    | 0.9 | 17,527.2 | 15,774.48 | 3943.62 | 11,830.86 |
| D3 | 0.7 | 32,420 | 22,694 | 5673.5 | 17,020.5 |
|    | 0.8 | 32,420 | 25,936 | 6484 | 19,452 |
|    | 0.9 | 32,420 | 29,178 | 7294.5 | 21,883.86 |
| N1 | 0.7 | 9676.6 | 6773.62 | 1693.41 | 5080.22 |
|    | 0.8 | 9676.6 | 7741.28 | 1935.32 | 5805.96 |
|    | 0.9 | 9676.6 | 8708.94 | 2177.24 | 6531.71 |
| N2 | 0.7 | 14,301.8 | 10,011.26 | 2502.82 | 7508.45 |
|    | 0.8 | 14,301.8 | 11,441.44 | 2860.36 | 8581.08 |
|    | 0.9 | 14,301.8 | 12,871.62 | 3217.91 | 9653.72 |
| N3 | 0.7 | 29,735.9 | 20,815.13 | 5203.78 | 15,611.35 |
|    | 0.8 | 29,735.9 | 23,788.72 | 5947.18 | 17,841.54 |
|    | 0.9 | 29,735.9 | 26,762.31 | 6690.58 | 20,071.73 |

Figure 2 shows how to set the device and the analyzed sample.

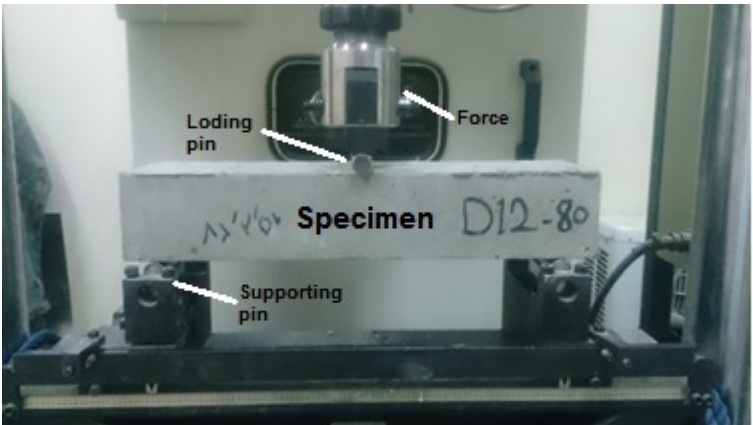

**Figure 2.** Test of fatigue life of concrete samples.

## 3. Results and Discussion

Table 7 presents the fatigue life of different concrete samples with three different thicknesses with and without fibers at three stress levels.

**Table 7.** Number of loadings of different fiber concrete samples.

| Specimen | S | N No. 1 | N No. 2 | N No. 3 | N Average | Percent Change with Respect to Plane Concrete (%) |
|---|---|---|---|---|---|---|
| D1 | 0.7 | 88,003 | 94,504 | 91,000 | 91,169 | 8.18 |
| | 0.8 | 11,502 | 12,488 | 12,014 | 12,001 | 29.14 |
| | 0.9 | 2018 | 1637 | 1916 | 1857 | 68.46 |
| D2 | 0.7 | 273,485 | 301,275 | 284,043 | 286,268 | 59.36 |
| | 0.8 | 17,683 | 17,560 | 17,496 | 17,580 | 43.97 |
| | 0.9 | 2086 | 2381 | 2201 | 2223 | 52.9 |
| D3 | 0.7 | 1,042,247 | 1,273,845 | 1,003,452 | 1,106,515 | 48.67 |
| | 0.8 | 32,870 | 29,381 | 28,947 | 30,399 | 24.53 |
| | 0.9 | 3367 | 3172 | 3045 | 3195 | 19.31 |
| N1 | 0.7 | 86,732 | 83,764 | 82,319 | 84,271 | 0 |
| | 0.8 | 9542 | 9126 | 9211 | 9293 | 0 |
| | 0.9 | 1040 | 1087 | 1180 | 1102 | 0 |
| N2 | 0.7 | 178,392 | 194,527 | 165,984 | 179,634 | 0 |
| | 0.8 | 12,634 | 11,996 | 12,005 | 12,211 | 0 |
| | 0.9 | 1317 | 1482 | 1562 | 1454 | 0 |
| N3 | 0.7 | 742,891 | 738,265 | 751,672 | 744,276 | 0 |
| | 0.8 | 23,671 | 24,678 | 24,882 | 24,410 | 0 |
| | 0.9 | 2543 | 2871 | 2619 | 2678 | 0 |

Considering the purpose of modelling, which was to evaluate the effect of fibers and sample size variations on the fatigue life of the concrete beam samples, and according to the literature review, the variables shown in Table 8 were used in constructing the model. The model used in this study was a multiple linear regression with the variables depicted in Table 8 as well as its common forms including double exponential, logarithmic and multiplication forms, in case of need.

**Table 8.** Variables used in modelling.

| Variable | Parameter | Unit |
|---|---|---|
| N | Number of loading | In cycle |
| S | Level of stress | - |
| H | Thickness of concrete specimen | Centimeter |
| Fiber dosage | Fiber content | Percent by volume |

Table 9 presents the general statistical data derived from the observations of the experiments for the given variables. The number of observations was 18.

**Table 9.** General information of observations for variables.

| Variable | Observation | Average | Deviation | Minimum | Maximum |
|---|---|---|---|---|---|
| N | 18 | 146,260.2 | 299,901.3 | 1402.333 | 1,106,515 |
| S | 18 | 0.8 | 0.084017 | 0.7 | 0.9 |
| H | 18 | 11 | 3.022927 | 8 | 15 |
| Fiber dosage | 18 | 0.2 | 0.205798 | 0 | 0.4 |

In the primary study of the data before modelling, it was observed that variable N with the correlation coefficient of $-0.5832$ had the highest correlation coefficient with S

among the independent variables. Thus, it was likely to have a significant effect on N. In the next ranking, variable H with a correlation coefficient of 0.4201 had the highest correlation with S.

For modelling, the independent variables S, H and Fiber dosage as well as their double exponential forms ($S^2$, $H^2$ and Fiber dosage$^2$) were considered. In addition, to investigate the mutual effect of independent variables, the multiplication forms of the variables (S.H, S. Fiber dosage and H. Fiber dosage) were considered in the modelling. Regarding the wide range of the dependent variable N as well as the literature review, the normal logarithmic form (Log(N)) of this variable was considered. After fitting the multiple regression model on the observations, the variables with no significant effect were removed from the model in successive steps. Subsequently, the final model was obtained, which is shown in Tables 10 and 11.

**Table 10.** ANOVA table.

|  | DF | SS | MS | F-Value | *p*-Value |
|---|---|---|---|---|---|
| Total | 17 | 78.32623 | 4.60742 | - | - |
| Model | 5 | 77.99716 | 15.59943 | 568.86 | 0.0000 |
| Residual error | 12 | 0.32907 | 0.02742 |  |  |
| $R^2$ | 0.9958 | - | - | - | - |
| Adjusted $R^2$ | 0.9940 | - | - | - | - |

**Table 11.** Values of regression coefficients for Log(N).

| Independent Variable | Regression Coefficient | *p*-Value | T-Value | Standard Error | Standardized Coefficients |
|---|---|---|---|---|---|
| Constant | 39.7292 | 0.000 | 7.29 | 5.45081 | - |
| S | −68.12385 | 0.000 | −5.09 | 13.37625 | −2.66646 |
| $S^2$ | 35.03476 | 0.001 | 4.23 | 8.279896 | 2.195526 |
| H | 1.068757 | 0.000 | 8.18 | 0.1305801 | 1.5083 |
| Fiber dosage | 0.63435 | 0.007 | 3.25 | 0.1951583 | 0.060819 |
| H S | −1.11486 | 0.000 | −6.87 | 0.1623815 | −1.353224 |

According to Table 10, the high F-value, or its equivalent low *p*-value, indicated the overall significance of the model at the confidence level of above 99%. Furthermore, since the initial objective of this modelling was to construct a powerful prediction model, the highness of the adjusted $R^2$ value was very important; accordingly, having a value of 0.9940 in this model indicated a very high capability of this model in prediction. The value of adjusted $R^2$ implied that the independent variables used in the model could justify 99.4% of the variations in the dependent variable, i.e., Log(N).

As demonstrated in Table 11, regarding the *p*-values, all three variables of S, H and Fiber dosage became significant at the confidence level of 99%. As for the stress level variable (S), its double exponential form became significant as well. Furthermore, this variable and H had a mutual effect, meaning that a change in one of them would influence the effect of these variables on the number of loadings. To investigate the effect of stress level variations on the number of loadings, Equation (1) was obtained through mathematical calculations and with regard to the resulting model. This equation %ΔN represents the percentage of variations in the number of loadings. Equation (1) indicates that the percentage of variations in the number of loadings due to the stress level variations depended on the initial stress level and height of the sample. For example, changing the stress level by 0.1% resulted in the variation percentage of the number of loadings, s in accordance with Figure 3. As can be seen, for such a change in stress level, the smaller the initial stress

level and the more the thickness of the sample, the higher the variation percentage of the number of loadings would be.

$$\frac{\%\Delta N}{100} \approx (-68.12385 + 70.694S - 1.11486H)\Delta S \tag{1}$$

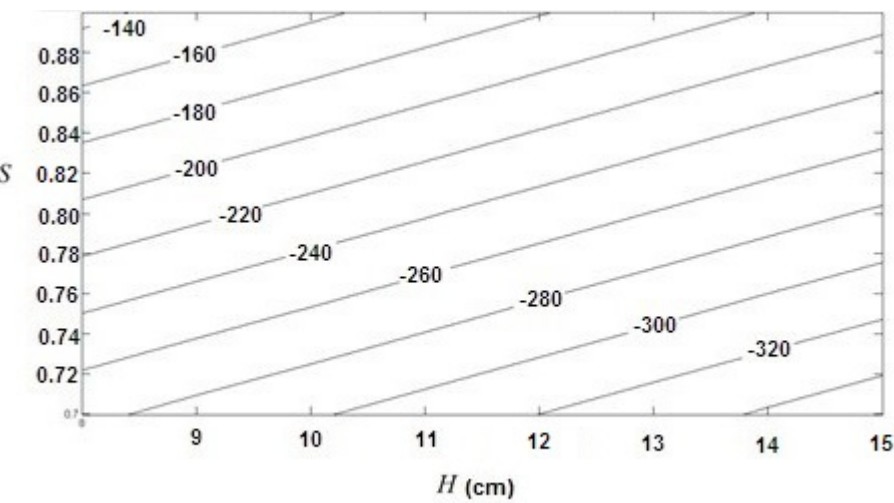

**Figure 3.** Variation percentage of N for variations of $\Delta S = 0.1$.

### 3.1. The Effect of the Sample Thickness

The effect of the sample's thickness variations on changes in the number of loadings was obtained in accordance with Equation (2), which indicated that the effect of thickness variations on the variation percentage of the number of loadings depended on the initial stress level. Figure 4 represents the effect of height variations at different stress levels on the variation percentage of stress level. As can be observed, for a certain change in height, the lower stress level would lead to lower variations in the number of loadings.

$$\frac{\%\Delta N}{100} \approx (1.068757 - 1.11486S)\Delta H \tag{2}$$

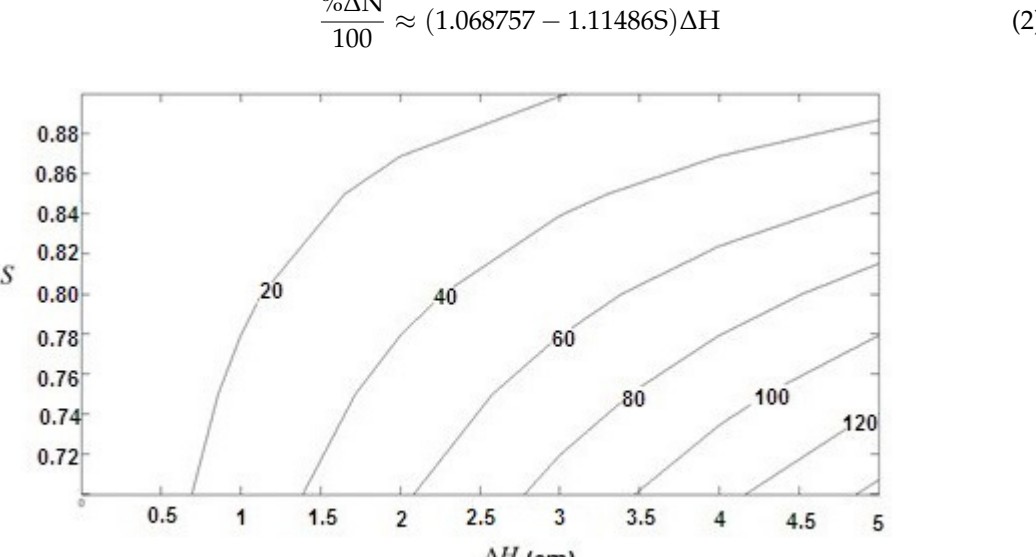

**Figure 4.** Variation percentage of the number of loadings for change in height at different stress levels.

### 3.2. The Effect of the Fiber Content

The effect of change in the fiber content percentage on the variation percentage of the number of loadings was obtained through Equation (3), which indicated a linear relationship between the change in fiber content percentage and variation percentage of the

number of loadings. As can be observed, per 0.1% increase in the fiber content percentage led to an increase by 6.3435 in the number of loadings. The relationship between the change in fiber content percentage and the variation percentage of the number of loadings is shown in Figure 5.

$$\frac{\%\Delta N}{100} \approx 0.63435 \Delta \text{Fiberdosage} \tag{3}$$

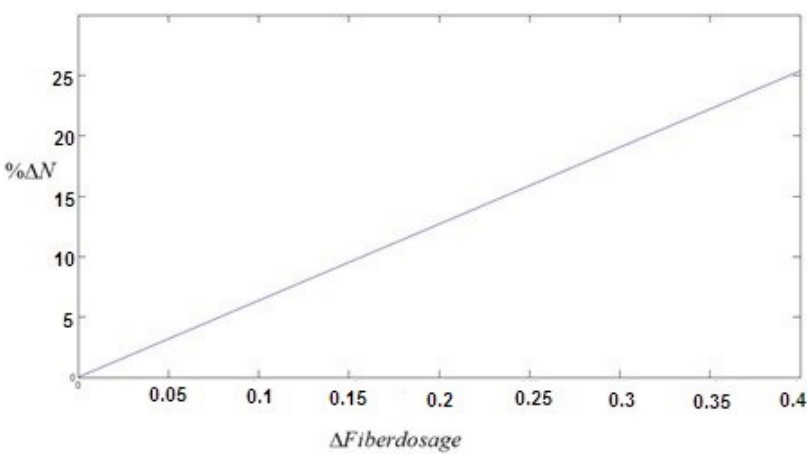

**Figure 5.** Effect of percent change in fiber on number of loadings.

To determine the variable with maximum effect on the loading percentage, the standardized coefficients were calculated for the variables, which are shown in the last column of Table 11. As can be observed, the largest effect was related to the stress level with a degree of two and then, the sample thickness. Figure 7 depicts the constructed model for the constant values of different variables as well as their upper and lower levels.

### 3.3. Assessment of Ruptured Cross-Section and Fibers

After the tests, the cross-sectional area of the sample was broken, and the tips of the fibers were examined.

Figure 6 shows a picture of broken fibers. A cross-sectional analysis of the broken sample showed that the failure of most tested concrete samples was initiated and extended from the aggregate and mortar parts. This indicates the transfer of load between cement mortar and aggregate and load bearing by aggregates of concrete. Thus, it can be stated that the concrete mix design was suitable. Additionally, according to Figure 6, it is seen that the fibers have not been ruptured due to elongation and also have not been pulled out which indicates good performance of the fibers.

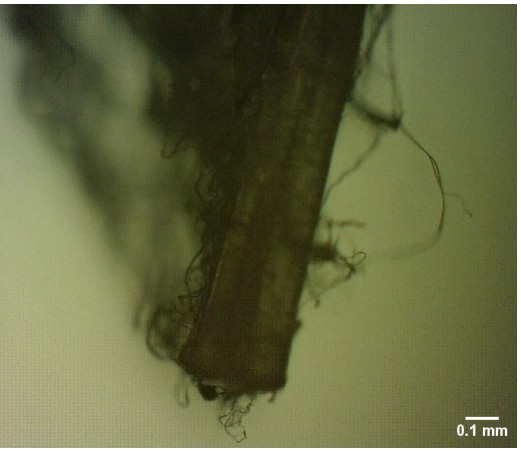

**Figure 6.** Zoomed view for one of the broken fibers.

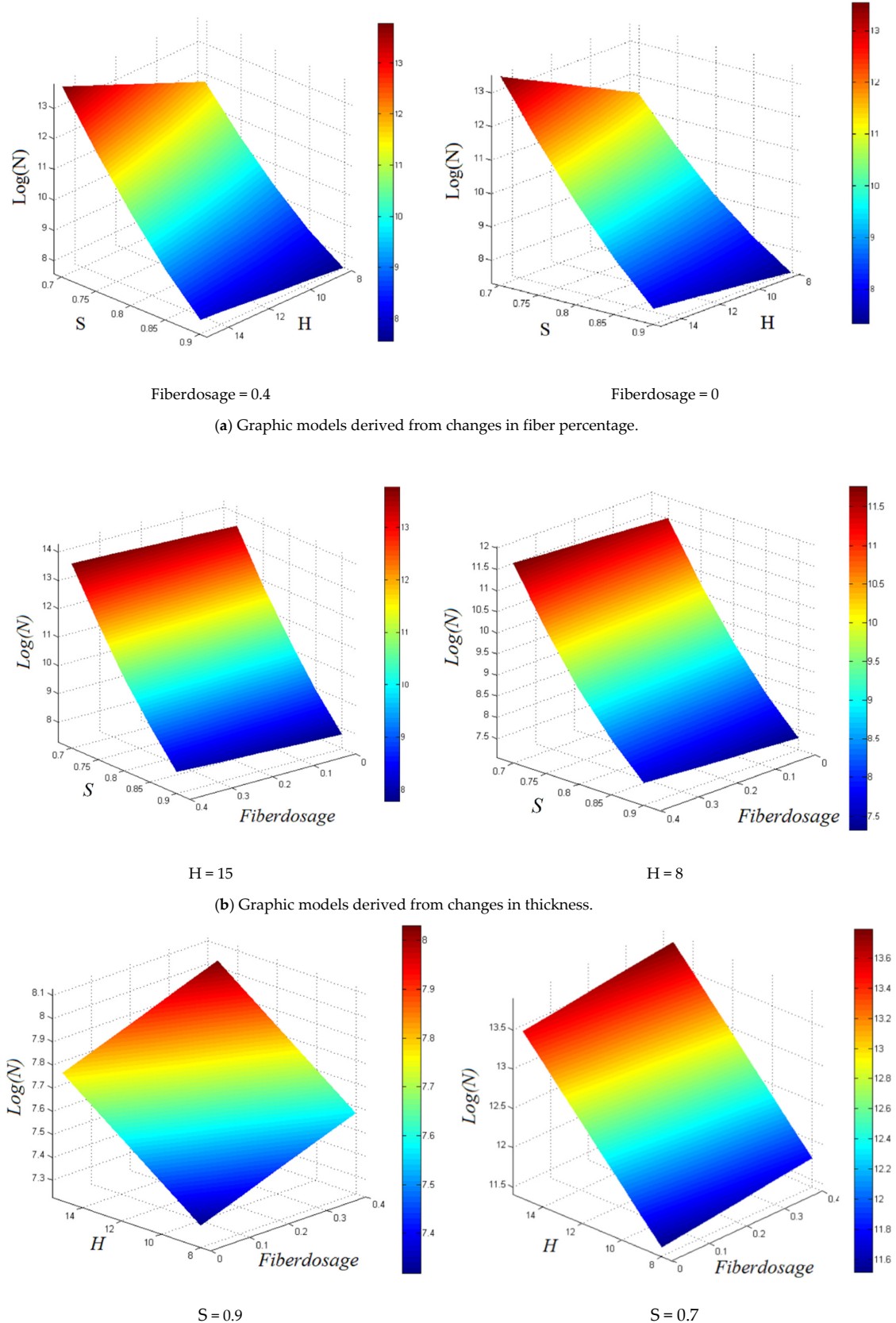

(**a**) Graphic models derived from changes in fiber percentage.

(**b**) Graphic models derived from changes in thickness.

(**c**) Graphic models derived from changes in stress level

**Figure 7.** Graphic models derived from changes in stress level, thickness and fiber percentage.

## 4. Conclusions

In the present research, fatigue life was measured for macro-synthetic fiber-reinforced concrete samples with the same mix design with three different thicknesses and the experimental model of the variations was provided. The results showed that increasing the thickness of the sample and adding fibers to concrete increases the bending strength and fatigue life. In other words, by adding fibers to concrete, the thickness of the concrete layer can be reduced to achieve a certain bending strength, which is effective in the implementation of concrete pavements and in places where there is a limit to the thickness of the concrete layer.

The main results of the present research are as follows:

- Results of the number of loadings of the concrete samples with different thicknesses show that with the addition of the macro-synthetic fibers to the concrete mixture, the number of loadings for the stress levels of 0.7, 0.8 and 0.9 was increased by 8.18–68.46%, 43.97–58.36% and 19.31–48.67%, respectively.
- According to the results, the intensity of degree-2 stress level and, additionally, thickness had the largest effects on the fatigue life of the concrete beam samples.

Finally, research on the effect of changing the width of the concrete beam on the fatigue life in order to simulate it with a concrete slab and research on the fatigue life of self-compacting, self-acting, polymer concrete, etc. are future research projects of the authors. Further, in the numerical modelling of crack propagation under flexural fatigue of concrete samples a fracture process zone will be included in the analysis by introducing microdamage models presented in [67–72].

**Author Contributions:** Conceptualization, M.D., A.H. and M.R.M.A.; methodology, M.D., A.H., M.R.M.A. and T.S.; validation, M.D. and A.H.; investigation, M.D. and A.K.; writing—original draft preparation, M.D., A.H. and M.R.M.A.; writing—review and editing, M.R.M.A. and T.S.; supervision, M.R.M.A. and T.S.; project administration, M.R.M.A. and T.S. All authors have read and agreed to the published version of the manuscript.

**Funding:** This research received no external funding.

**Data Availability Statement:** Not applicable.

**Conflicts of Interest:** The authors declare no conflict of interest.

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
