# Peer review of "Experimental Model for Study of Thickness Effect on Flexural Fatigue Life of Macro-Synthetic-Fiber-Reinforced Concretes"

_buildings, doi:10.3390/buildings13030642_

Round 1
Reviewer 1 Report (New Reviewer)
This work is very confusing, and the writing is so casual that the reviewer fail to understand the meaning and merit of this work. There are so many questions, I can only list a few for the authors’ reference.
1. Why the authors highlight some many text in yellow? The writing here is so casual that it does not look like a scientific paper.
2. What in earth did the authors add to the concrete? 4% steel fibers (as per the Abstract) or 0.4% steel fibers (as per Table 3 and Section 2.1)?
3. What specimens did the authors test, E1,E2,E3 as per Table 3 or D1,D2,D3 as per Table 4 and 5?
4. I guess the counterpart of N1 is D1 in Table 5, I guess N is the numbers, S is the stress range in Table 5. Then, for S=0.7, the average N of N1 is 84271.67, the average N of D1 is 91169.00. It only increases 8% from N1 to D1, then how is the 30.49% change come from?
Author Response
Comment: Why the authors highlight some many text in yellow?
Answer: checked and corrected.
Comment: What in earth did the authors add to the concrete? 4% steel fibers (as per the Abstract) or 0.4% steel fibers (as per Table 3 and Section 2.1)?
Answer: As mentioned in section 2.1, superplasticizer 2.2 (kg/m3), and fibers with dose of 3.6 (kg/m3) are added to the concrete mixture.
Comment : What specimens did the authors test, E1,E2,E3 as per Table 3 or D1,D2,D3 as per Table 4 and 5?
Answer: checked and corrected.
Comment : I guess the counterpart of N1 is D1 in Table 5, I guess N is the numbers, S is the stress range in Table 5. Then, for S=0.7, the average N of N1 is 84271.67, the average N of D1 is 91169.00. It only increases 8% from N1 to D1, then how is the 30.49% change come from?
Answer: checked and corrected.
Comment: Why was 0.4% volume percentage of twisted fibres selected?
Answer: The purpose of this research is to provide an experimental model of concrete sample thickness change with fatigue life. Based on the literature review and past researches of the authors and to limit the number of experiments, the amount of fibers was chosen to be 0.4% which according to the research method will not have an effect on the presented models.
Comment: Section 2 is very weak and it should be reframed completely.
Answer: checked and corrected.
Comment: How were the results from Table 4 obtained?
Answer: X-Ray Fluorescence Spectrometer (XRF)
Comment: The discussion section is shallow and it is recommended to add more discussion from each figure and compare the results with the literature.
Answer: checked and corrected.
Comment: Please revise the conclusions based on the new discussions.
Answer: checked and corrected.

Reviewer 2 Report (New Reviewer)
The purpose of this paper is to examine how changing the specimen size affects the flexural fatigue life of macro-synthetic-fibre-reinforced concretes. Most section of this article is not written well and it requires extensive improvements.
1. Start the abstract by placing the question of current issues in this research area, the purpose of the study, the objective, the methodology adopted and the results. The current form of abstract is lacking in all the aspects mentioned above. It is highly recommended to rewrite the abstract.
2. What is the novelty of this research?. Please add this at the end of the introduction.
3. There are no proper details and discussion about the materials used, specimen preparation, and mixture id.
4. The mixing combination should be presented in Table form.
5. Why was 0.4% volume percentage of twisted fibres selected?
6. Section 2 is very weak and it should be reframed completely.
7. How were the results from Table 4 obtained?
8. The discussion section is shallow and it is recommended to add more discussion from each figure and compare the results with the literature.
9. Please revise the conclusions based on the new discussions.
Author Response
Comment: Start the abstract by placing the question of current issues in this research area, the purpose of the study, the objective, the methodology adopted and the results. The current form of abstract is lacking in all the aspects mentioned above. It is highly recommended to rewrite the abstract.
Answer: checked and corrected.
Comment: What is the novelty of this research?. Please add this at the end of the introduction.
Answer: As mentioned at the end of the introduction, in most of the studies, the concrete sample's thickness is increased along with the increase in the beam's length which, according to the authors, does not provide an accurate assessment of the effect of sample size on fatigue life; however, in the present work, only the thickness of the beam samples with and without fibres was changed and other dimensions of the samples were kept constant to investigate merely the effect of increased thickness. The purpose of this research is to provide an experimental model of concrete sample thickness change with fatigue life, which is different from other researchers in the form and purpose of the research
Comment: There are no proper details and discussion about the materials used, specimen preparation, and mixture id.
Answer: checked and corrected.
Comment: The mixing combination should be presented in Table form.
Answer: checked and corrected.
Comment: Why was 0.4% volume percentage of twisted fibres selected?
Answer: This research aims to provide an experimental model of concrete sample thickness change with fatigue life. Based on the literature review and past research of the authors and to limit the number of experiments, the number of fibres was chosen to be 0.4% which according to the research method will not have an effect on the presented models.
Comment: Section 2 is very weak and it should be reframed completely.
Answer: checked and corrected.
Comment: How were the results from Table 4 obtained?
Answer: X-Ray Fluorescence Spectrometer (XRF)
Comment: The discussion section is shallow and it is recommended to add more discussion from each figure and compare the results with the literature.
Answer: checked and corrected.
Comment: Please revise the conclusions based on the new discussions.
Answer: checked and corrected.

Reviewer 3 Report (New Reviewer)
1 The abstract must have a background of the research, onjective, methodology, findings and concluding remarks with minimum 250
2 In results are just mentioned as it is, it must be discussed as, how, why you get that result, what influenced to get this result....
3 Dont write about table in the conclusion
4 Figure 5 must have 0.15,0.2 in the x axis
5 Figure 4 , write the y axis title, similar check in all images
6 Separate the table 2a and 2b as, 2 and 3
7 Some sentences are too long. Generally, it is better to write short sentences with one idea per sentence.
8 Contributions should be highlighted more. It should be made clear what is novel and how it addresses the limitations of prior work.
9 The authors should explain clearly what the differences are between the prior work and the solution presented in this paper.
10 The experiments should be updated to include some comparison with newer studies.
11 There is not enough discussion of the experimental results.
12 Some text must be added to discuss the future work or research opportunities.
Author Response
Reviewer 4
1 The abstract must have a background of the research, onjective, methodology, findings and concluding remarks with minimum 250
Answer: checked and corrected.
2 In results are just mentioned as it is, it must be discussed as, how, why you get that result, what influenced to get this result....
Answer: checked and corrected.
3 Dont write about table in the conclusion
Answer: checked and corrected.
4 Figure 5 must have 0.15,0.2 in the x axis
Answer: checked and corrected.
5 Figure 4 , write the y axis title, similar check in all images
Answer: checked and corrected.
6 Separate the table 2a and 2b as, 2 and 3
Answer: it was corrected.
7 Some sentences are too long. Generally, it is better to write short sentences with one idea per sentence.
Answer: the English language was corrected by the native language speaker.
8 Contributions should be highlighted more. It should be made clear what is novel and how it addresses the limitations of prior work.
Answer: the novel was more precise and a comparison to the previous articles was done.
9 The authors should explain clearly what the differences are between the prior work and the solution presented in this paper.
Answer: explanations were done
10 The experiments should be updated to include some comparison with newer studies.
Answer: checked and corrected.
11 There is not enough discussion of the experimental results.
Answer: checked and corrected.
12 Some text must be added to discuss the future work or research opportunities.
Answer: checked and corrected.

Round 2
Reviewer 1 Report (New Reviewer)
The paper has been improved.
However, the introduction is still a little weak, the authors should add some reviews on the steel fibers on the static bending behavior of concrete beams (Engineering Structures 272 (2022) 115020). And then discuss its dynamic fatigue behavior. In this way, the transition is smoother, and easier for readers to follow.
In addition, the authors should double-check the grammar, typos, and format of the tables. They are still far from standard.
Author Response
Dear Reviewers
Authors would like to thank reviewers for the time they dedicated to review the paper. Here are the responses to the issues raised. The parts that have been added to the text are shown in highlighted color.
Reviewer 1:
The paper has been improved. However, the introduction is still a little weak, the authors should add some reviews on the steel fibres on the static bending behaviour of concrete beams (Engineering Structures 272 (2022) 115020). And then discuss its dynamic fatigue behaviour. In this way, the transition is smoother and easier for readers to follow. In addition, the authors should double-check the grammar, typos, and format of the tables. They are still far from standard.
Answer: some other related and relevant papers were reviewed and cited in the literature review. The grammar and format of tables and figures were also checked and corrected.

Reviewer 2 Report (New Reviewer)
Response to the reviewer is sufficient.
Author Response
Reviewer 2
Comment: Response to the reviewer is sufficient.

Reviewer 3 Report (New Reviewer)
Dear Author,
1. Table 10 is in different format
2. Cnage the figure 7 caption, its a broken size image or cross sectional area ?
Author Response
Reviewer 3
Comment: Table 10 is in different format.
Answer: checked and corrected.
Comment: Change the figure 7 caption, its a broken size image or cross sectional area ?
Answer: the caption of this figure was revised and clarified.

This manuscript is a resubmission of an earlier submission. The following is a list of the peer review reports and author responses from that submission.
Round 1
Reviewer 1 Report
Dear authors,
you have collect data and present an interesting work. In my opinion this manuscript is suitable for this journal after a moderate revision:
The abstract is too brief
Please add more references eg what about the use of sustainable fibres in concrete?
Please check your text for editing corrections eg In Table 4 there is no the (a) and (b)
Please add scale in Fig 1, 7
Please add some compositional data about your aggregates if you have
What about microstructural characteristis and images from concrete texture. Please add or refer something. Moreover, more discussion about your results is needed,:eg "failure was from aggregate..." please explain
Good luck
Kind regards
Reviewer 2 Report
The submitted research paper “buildings-2147459” entitled: “Experimental model for the effect of specimen size on Flexural Fatigue Life of macro-synthetic-fiber-reinforced concretes” is an experimental investigation. The authors studied the influence of addition of macro-synthetic fibers to the fatigue life of concrete samples and additional the size effect of the specimens. In general, the submitted Article is very weak for publication.
· The abstract is very short and fairly informative
· The state-of-the-art is presented in a confusing way. The Introduction is not well structured. The research gap is not presented and nor is the significance of the current research.
· The manuscript has also serious flaws and the overall quality, including value of contribution, methodology and presentation style is very low for a typical journal paper. I found the text difficult to understand.
· The materials and methods are barely described as well as the test set up and instrumentation.
· The experimental program is not presented in detail.
· Too many tables are included in the manuscript and they are not well organised and presented as well.
· There is no discussion within the manuscript, there are no explanations. The authors just included a bunch of tables and did not accompany with any text. There are 15 pages of manuscript of which two are introduction, one is reference, and almost 10 are figures and tables, there is no commenting and discussion, this cannot be considered a research manuscript.
· The representation of the results in not clear. The manuscript is hard to follow and I cannot spot the significance of the research.
I believe that the authors have rushed a bit to submit the manuscript while they should give more time to their research and further develop their idea. The central idea of the manuscript seems to be interesting but the authors have to better present and explain their work so their efforts are recognized. The discussion section and especially the conclusions section needs to be enriched. For these reasons, I cannot consider the paper suitable for publication. I suggest authors to strongly review their paper, further develop their research and to re-submit a new, revised and clean version of the manuscript.
Reviewer 3 Report
The article is devoted to the study of fatigue life of samples from macro-synthetic-fiber-reinforced concretes. A large number of studies have previously been carried out in this area, which is confirmed by the literature review given in the article. As a novelty of the study, the authors position the use of sample thickness as a variable factor. In my opinion, the design of the study looks rather strange. It is well known that the bending strength of beams is directly proportional to their thickness. As a result of the study, the authors also get the result that the number of cycles is directly proportional to the thickness of the beam (formula 2).
Therefore, I think that this article is not of great interest and does not deserve publication in the journal Buildings.